# Risk Assessment of Perioperative Respiratory Adverse Events and Validation of the COLDS Score in Children with Upper Respiratory Tract Infection

**DOI:** 10.3390/medicina58101340

**Published:** 2022-09-23

**Authors:** Hyo Sung Kim, Young Sung Kim, Byung Gun Lim, Jae Hak Lee, Jihyun Song, Heezoo Kim

**Affiliations:** Department of Anesthesiology and Pain Medicine, Korea University Guro Hospital, Korea University College of Medicine, Seoul 08308, Korea

**Keywords:** COLDS score, pediatric anesthesia, perioperative respiratory adverse events, pre-anesthetic risk assessment, upper respiratory tract infection

## Abstract

*Background**and**objectives:* Children are at greater risk of upper respiratory tract infection (URTI), which can pose a higher risk of perioperative respiratory adverse events (PRAEs), than adults. The purpose of this study was to validate the COLDS score as a pre-anesthetic risk assessment tool for predicting the possibility of PRAEs. *Materials and methods:* Children aged under 18 years and undergoing elective surgery were retrospectively included. Logistic regression analysis and the area under the receiver-operating characteristic (ROC) curve (AUC) were used to estimate the ability of the COLDS score to predict PRAEs. Propensity-matched comparison was evaluated using the cut-off value from the ROC curve. *Results:* Among the 6252 children, 158 children had a recent URTI and 34 cases of PRAEs were reported. Age, current symptoms, and COLDS score were found to be significant variables in predicting PRAEs. From the ROC curve, values of 0.652 (*p* = 0.007) for AUC and 12.5 for the cut-off value of the COLDS score were calculated. Propensity-matched comparison revealed that each and every component of COLDS contributed to the higher COLDS score group. In addition to higher COLDS score, younger age and current URTI symptoms were found to be significant risk factors for PRAEs. *Conclusions:* This study validated the predictive power of COLDS score as a risk assessment tool for children with URTI undergoing elective surgery under general anesthesia.

## 1. Introduction

Children with current or recent upper respiratory tract infection (URTI) have an increased risk of perioperative respiratory adverse events (PRAEs), including cough, laryngospasm, bronchospasm, and desaturation [1,2,3]. PRAEs account for 27% of perioperative cardiac arrest occurrences in pediatric surgery [4]. Anesthetic management of patients with PRAEs has improved; however, the possibility of a catastrophic outcome of PRAEs remains a major reason for morbidity and mortality during pediatric anesthesia [5].

URTI symptoms include nasal congestion, rhinorrhea, cough, sore throat, fever, and malaise [6]. They are more common in children than in adults. Their incidence increases at younger ages. Children aged ≤4 years suffer an average of eight colds per year, and their chances of experiencing colds decrease as they get older [6]. The symptoms usually last for 7 to 10 days, but hypersensitivity of the airway can persist for approximately 6 weeks [6,7]. Because hypersensitivity of the airway increases the risk of PRAEs, children with a current or recent URTI are recommended to postpone surgery. In previous research, the incidence of PRAEs in children who had a current or recent URTI was shown to be 25–29%, whereas that in children without a URTI was 12% [1,8]. Zhang et al. [9] also reported that children who had a URTI within 2 weeks of surgery showed a significantly higher incidence of PRAEs. Therefore, when children have a severe URTI, delaying surgery by at least 2 weeks after active URTI symptoms are present is desirable [9,10]. In cases of URTI onset within 2–6 weeks of surgery, the risk of airway hypersensitivity may remain. However, elective surgery may proceed considering the benefit of surgery. Postponing surgery can be stressful not only for the child and the parents but also for the surgeon and the hospital. Moreover, in emergency surgery, the risk of proceeding with surgery can outweigh the risk of PRAEs.

The precise risk prediction of PRAEs in children with URTI symptoms is critical in deciding whether to postpone surgery and in optimizing perioperative management. Determining the risk of PRAEs can reduce emotional damage to patients and financial damage to both patients and the hospital with regard to delays in surgery. Several risk factors for PRAEs have been identified. The following patient-related factors influence PRAE occurrence: age ≤ 6 years; URTI symptoms (respiratory symptoms such as purulent secretion, moist cough, nasal congestion, etc.); primary pulmonary morbidity (respiratory syncytial virus infection, asthma, prematurity, bronchopulmonary dysplasia, cystic fibrosis, pulmonary hypertension, etc.); infectious disease; fever ≥38.5 °C; exposure to secondhand smoke; parental mention of the child’s symptoms [11,12,13]. Anesthetic factors which influence PRAE occurrence include endotracheal intubation, anesthetic agents, and anesthesiologist experience [11]. Surgical-related factors, such as airway surgery, ear–nose–throat surgery, and eye surgery, are also known to contribute to the occurrence of PRAEs [7].

The COLDS score is a pre-anesthetic scoring system which takes into account risk factors for PRAEs to assess the risk of PRAEs in children with URTI presenting for elective surgery [14]. It is composed of five categories, and the COLDS acronym stands for each of the categories (Appendix A). Each category is assigned 1, 2, or 5 points according to the degree to which it affects the occurrence of PRAEs. “C” stands for current symptoms, including congestion, cough, fever, and sputum. Children without URTI symptoms are given 1 point, whereas those with symptoms of congestion, clear rhinorrhea, sore throat, sneezing, low fever, or dry cough are deemed to have a mild risk for PRAEs and are given 2 points. Purulent sputum, wet cough, and high fever are considered moderate to severe risk symptoms, and 5 points are given to children who have these symptoms. “O” is the onset of URTI symptoms. If a child experienced URTI symptoms 4 weeks before the date of surgery, only 1 point is given, but 2 or 5 points are given if a child experienced URTI symptoms 2–4 weeks before the date of surgery or within 2 weeks of surgery, respectively. “L” stands for lung disease, including those conditions that can increase airway hypersensitivity. Children without lung disease are given 1 point. Children with intermittent asthma, or a history of respiratory syncytial virus infection or passive smoking are given 2 points, and children with persistent asthma, pulmonary hypertension, or chronic lung disease are given 5 points. “D” stands for the airway device. Perioperative use of a facemask is the safest approach with less irritation to the airway during surgery, and the use of a laryngeal mask has been shown to reduce the occurrence of cough [15,16]. Therefore, 1 point is given to children who undergo surgery with a facemask and 2 points are given to those who use a laryngeal mask airway or other supraglottic airway device during surgery. Five points are given to children who undergo tracheal intubation, which is considered risky compared with other airway devices. “S” stands for surgery type. Children who undergo tonsillectomy, adenoidectomy, or flexible bronchoscopy, which are considered minor risk surgeries, are given 2 points. Those who undergo cleft palate repair, rigid bronchoscopy, or maxillofacial surgery, which are classified as moderate to severe risk surgeries, are given 5 points. Children who undergo other types of surgery which are not mentioned above are given 1 point. Overall, the total score of the five categories ranges from 5 to 25. As the score increases to 25, the likelihood of PRAE occurrence increases [14,17].

The purpose of this study was to assess the predictive value of this risk stratification scheme and to determine whether a high COLDS score indicates a higher occurrence of PRAEs. By demonstrating the validity of the COLDS score, the risk of PRAEs can be better anticipated and PRAE management can be made easier for anesthesiologists.

## 2. Materials and Methods

### Data Collection

In this single-center retrospective study, 6252 of the 7468 patients selected aged <18 years underwent elective surgery under general anesthesia from January 2013 to December 2018 at Korea University Guro Hospital. Demographic information, such as age, sex, height, and weight; peri-operative image findings; laboratory data; and peri-operative URTI symptoms were collected. Data on perioperative URTI symptoms were obtained from the medical records and preoperative summary of each patient. Since 2013, the in-department standards for URTI-related medical records have been strengthened, and postoperative respiratory events have been described in detail. The determination of PRAE was achieved by the presence of persistent cough, breath holding, hypoxemia (oxygen saturation < 95%, lasts more than 30 s), laryngospasm, or bronchospasm. Perioperative follow-up data were evaluated to confirm the occurrence of PRAEs using intraoperative anesthetic records and postoperative medical records in the postoperative anesthetic unit (PACU) and ward. As mentioned above, patients with current or recent URTIs had their respiratory symptoms and progression of symptoms in the intraoperative and postoperative period documented in detail.

Patients undergoing emergency surgery were excluded from the study. Data relating to preoperative URTI symptoms were divided into the following five categories: URTI duration; URTI termination; presence of active URTI; URTI symptoms (nasal congestion, rhinorrhea, cough, and fever); chest X-ray findings compatible with URTI or pneumonia. The preoperative summary also included underlying diseases of the patient, asthma or allergy, and passive smoking history. Surgery-related data, such as type of surgery and duration of surgery, were also summarized. The total COLDS score was calculated based on these data. Postoperative information included duration of stay in the post-anesthetic care unit; postoperative symptoms; and medications, including opioid, steroid, or antibiotic prescriptions.

Logistic regression analysis was used to assess the PRAEs prediction of the COLDS score and other independent variables, including age, sex, weight, presence of ongoing URTI symptoms, and anesthetic time. The backward LR method was used to eliminate insignificant variables. The receiver operating characteristic (ROC) curve was drawn from the logistic regression analysis to assess the ability of the COLDS score to predict PRAEs, and the area under the ROC curve (AUC) was analyzed. The cut-off value that minimized (1-sensitivity)^2^ + (1-specificity)^2^ was calculated from the ROC curve, and two groups were then divided based on cut-off value as follows: low (group L) and high (group H) COLDS groups. To minimize the baseline imbalance of candidate variables between the two groups, we used propensity score matching and compared the outcome variables between the propensity-matched groups.

## 3. Results

From 2013 to 2018, the perioperative data of 7486 patients were analyzed. Of these, 6252 underwent elective surgery. A total of 158 patients had a current or recent URTI, and 34 cases (21.5%) of PRAEs were reported among the pediatric patients with a current or recent URTI. Fourteen patients displayed persistent cough, ten displayed breath holding, twenty-seven displayed hypoxemia, and two displayed laryngospasm. Seven patients used steroid or inhaler in the PACU. One patient showed newly occurred pneumonic infiltration in the postoperative chest X-ray. The distribution of the number of patients according to COLDS score is presented in Figure 1a. The maximum peak was shown at a COLDS score of 10, and the second peak was shown at 14. The number of patients with a score of 12 was relatively small (Figure 1a). Among the five components of COLDS, the score for “D” was significantly higher than that for the other four components. “O” also showed a significantly higher score than “C”, “L”, or “S” (Figure 1b).

Logistic regression analysis revealed that age (*p* = 0.003), ongoing URTI symptoms (*p* = 0.024), and COLDS score (*p* = 0.028) were significant variables in the final model for predicting PRAEs (Table 1). The initial logistic regression model is presented in Appendix A. The ROC curve was drawn and the estimated AUC was 0.652 (*p* = 0.007). The cut-off value to minimize (1-sensitivity)^2^ + (1-specificity)^2^ was 12.5 (Figure 2).

To understand the demographic information of the participants, patients were divided into low (group L) and high (group H) COLDS groups using a cut-off value of 12.5. Group L contained 89 patients and group H contained 69 patients (Appendix A). To match the baseline difference in candidate variables, propensity score matching was performed (Table 2), and each group contained 69 patients. Sputum was more frequent in patients from group H (*p* = 0.020), whereas rhinorrhea was more frequent in patients from group L (*p* = 0.044). The incidences of cough (*p* = 0.133) and fever (*p* = 0.980) were not meaningfully different between patients from the two groups. Active URTI symptoms on the day of surgery were more frequent in patients from the high COLDS score group (*p* < 0.001) than in the low COLDS score group. All patients in group L had normal chest X-ray findings, and eight patients in group H had abnormal chest X-ray findings (*p* = 0.006). The duration of URTI was comparable between the two groups (*p* = 0.450). However, URTI onset was closer to the day of surgery in group H (*p* < 0.001).

Components of the COLDS score were compared between the two groups. Mean scores of the patients in groups L and H in “C”, “O”, “L”, “D”, and “S” were 1.22 ± 0.42 and 1.68 ± 0.96; 1.91 ± 0.78 and 4.43 ± 1.25; 1.01 ± 0.12 and 1.68 ± 1.47; 4.58 ± 1.18 and 5 ± 0; and 1.09 ± 0.28 and 1.45 ± 1.14, respectively. Overall, all the components of the COLDS score showed a significantly higher number in group H (*p* < 0.001 in “C”, “O”, “L”; 0.004 in “D”; and 0.013 in “S”) than in group L. The total COLDS score was 9.80 ± 1.38 in group L and 14.25 ± 1.55 in group H (*p* < 0.001), showing a higher total COLDS score in group H than in group L.

## 4. Discussion

Despite its importance and necessity, there is no gold standard for preoperative risk assessment tools for PRAEs. Subramanyam et al. [18] evaluated a PRAE risk prediction tool with an AUC of 0.63–0.64 for pediatric ambulatory anesthesia. They proposed risk factors including age under 3 years (1 point), ASA physical status II and III (1 and 2 points respectively), morbid obesity (2 points), pre-existing pulmonary disorder (2 points), and surgery (3 points) versus radiology. Although they proposed an optimal cut-off of 4 (according to their risk score, maximum 10) for PRAE prediction, it was difficult to apply in our center. For example, according to their classification, all children under 3 years undergoing any surgery (minimum 4 points) are considered as high risk for PRAEs. Their PRAE incidence (2.8%) was quite low compared to ours (21.5%) because of the difference in patients’ characteristics.

In addition to the risk factors considered in this study, various risk factors including passive smoking, snoring, induction agents, and reversal agents have been proposed [19] but it has been difficult to design an appropriate risk prediction tool. One reason is that patient characteristics are different at each center, but even more problematic is the fact that these risk factors are often not independent of each other. While applying COLDS in our clinical practice, we considered the characteristics of the COLDS scoring system including five independent categories and equal distribution of risk weights.

The results of logistic regression analysis suggested that younger age and current symptoms affected the incidence of PRAEs. Moreover, a high COLDS score increased the risk of PRAEs. Previous studies have suggested numerous risk factors for PRAEs in children [11,12,20]. Consistent with their findings, we confirmed that children with younger age and current URTI symptoms are at increased risk of PRAEs when undergoing scheduled surgery under general anesthesia. We also confirmed that a higher COLDS score indicates a higher occurrence of PRAEs, and that COLDS score can be used as a simplified PRAEs risk calculating tool. Our findings have value in that they not only look plausible, but also present specific statistical values.

After the logistic regression analysis of the COLDS score and occurrence of PRAEs, the estimated AUC value was 0.652. This is slightly lower than in a prior study validating the COLDS score [17]. According to their study, the AUC value of the COLDS score for predicting PRAEs was 0.69. They also reported AUCs by age sub-groups (AUCs of ages 0–2 years, 2–4 years, and 4–6 years were 0.70, 0.71, and 0.66, respectively) [17]. Because they limited the ages of included patients, it is assumed that the AUCs for PRAEs were higher than ours. Meanwhile, considering that tools with an AUC of over 0.7 have a moderate diagnostic ability, the AUC found in the current study might not be adequate as a diagnostic tool but sufficient as a predictive tool. In addition, the COLDS score requires only the preoperative condition of the patients and does not need additional evaluation. Thus, it is a potentially useful and easy tool to predict PRAEs.

Previous studies have reported that URTI symptom severity can increase the occurrence of PRAEs in patients [20]. In our study, the percentage of patients with PRAEs increased as the COLDS score increased, which indicates that patients with a high COLDS score have a high risk of PRAEs. The analysis of demographic data showed important findings. First, considering the onset of URTI, patients who experienced a URTI > 1 week before the date of surgery were more prevalent in group L, whereas group H included more patients with a URTI within 1 week from the date of surgery. In particular, active URTI was more frequent in group H. This result accorded with that observed in previous studies stating the necessity of delaying surgery by 1 week for patients with an active URTI [9,10]. All five components of the COLDS score were higher in group H. This indicates that the COLDS score has a discriminative capacity regarding underlying risk factors of patients. With regard to the symptoms of URTI, rhinorrhea was more frequent in group L, whereas sputum was more prevalent in group H. This means that patients with purulent sputum are at more risk of PRAEs than patients with rhinorrhea. In addition, preoperative abnormal chest X-ray findings were found only in patients in group H, indicating that an abnormal chest X-ray finding increases the occurrence of PRAEs.

This study has several limitations. First, given the retrospective nature of the study, concerns with documented data exist. In particular, descriptions of the severity of URTI and PRAE symptoms can vary between clinicians. A large observational study by Michel et al. [20] and other previous studies have reported incidences of PRAEs of approximately 30% in pediatric patients with URTI, which was higher than that found in our study (21.5%). This may be due to differences in patient characteristics, but ambiguity in the URTI or PRAE definitions may also be a cause. As described in the Methods section, the documentation was based on hospital guidelines on respiratory symptoms. Because our hospital guidelines emphasized URTI experience, especially in pediatric patients, errors in data interpretation would be minimized among clinicians before discharging from the PACU. However, there still remains a possibility of insufficient information regarding PRAEs after PACU discharging. Second, the number of patients with an active URTI was relatively low because surgeries were commonly postponed in children with these symptoms. Moreover, where active URTI was recorded in the preoperative summary, the anesthetic clinician took steps to reduce the incidence of PRAEs. Therefore, the occurrence of PRAEs could have been reduced. If more supplemental data regarding active URTI were available, the AUC of the COLDS score might be higher than the result of the current study. However, owing to ethical issues, elucidating the validation of the COLDS score with active and especially severe URTI symptoms might be difficult.

Nevertheless, this study has clinical significance. Since the data of this study reflect actual clinical trials, similar results can be expected from other centers. When considering the delay period of surgery in children with URTI in clinical practice, it can be helpful to establish a standard based on the data in this study. In addition, this study was able to show the structural limitations of the COLDS scoring system. Each component of COLDS did not spread evenly, and in fact, it was found that “O” and “D” had to be higher than the other three components. In addition, the fact that 3 or 4 points are not assigned to each component is considered to be a limitation in that the COLDS score cannot be drawn continuously as a normal distribution curve.

Overall, this study supports the use of the COLDS score as a predictive tool for the occurrence of PRAEs. As the study collected retrospective data, it included a broad variety of patients. It encompasses various types of surgery, patients, and symptoms. Because young age affects the prevalence of PRAEs, if the study participants included a higher number of young patients, a higher predictive value would be obtained. Furthermore, although a previous study emphasized that age of <6 years affects the occurrence of PRAEs [8], our study showed the possibility of broader applications of the COLDS score regarding age. Other factors, such as anesthesiologist proficiency and anesthetic agents, also affect the occurrence of PRAEs [20,21]. Therefore, with more studies regarding other factors, an improved COLDS score tool could be developed.

## 5. Conclusions

In conclusion, the COLDS score is potentially helpful in decision making on whether to delay surgery and in assisting the anesthetic management of children. Through this simple guidance, the burden not only on patients and family members but also on medical teams can be lessened. Furthermore, as it is an easy and meaningful tool, its application can be broadened; for example, it can be applied to the sedation of outpatients. More investigations focusing on children with URTI symptoms should be performed in the future.

## Figures and Tables

**Figure 1 medicina-58-01340-f001:**
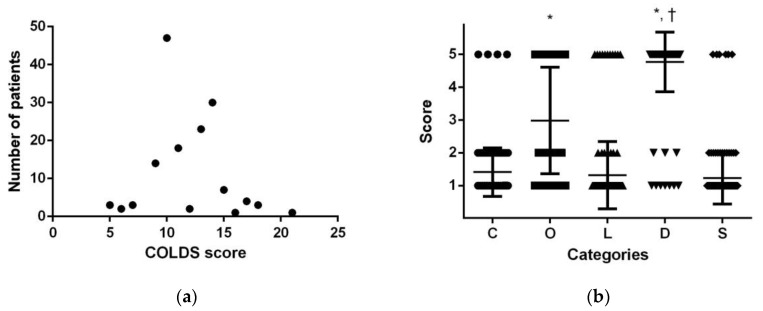
Graphs depicting the occurrence of perioperative respiratory adverse events (PRAEs) according to COLDS score. (**a**) Number of patients who suffered PRAEs according to COLDS score. (**b**) Description of each COLDS category in the patients with PRAEs. The five components of the COLDS score did not contribute equally to the total score. Total COLDS scores were contributed in the order of “D”, “O”, and the others. * *p* < 0.05 compared to “C”, “L”, or “S”, † *p* < 0.05 compared to “O”.

**Figure 2 medicina-58-01340-f002:**
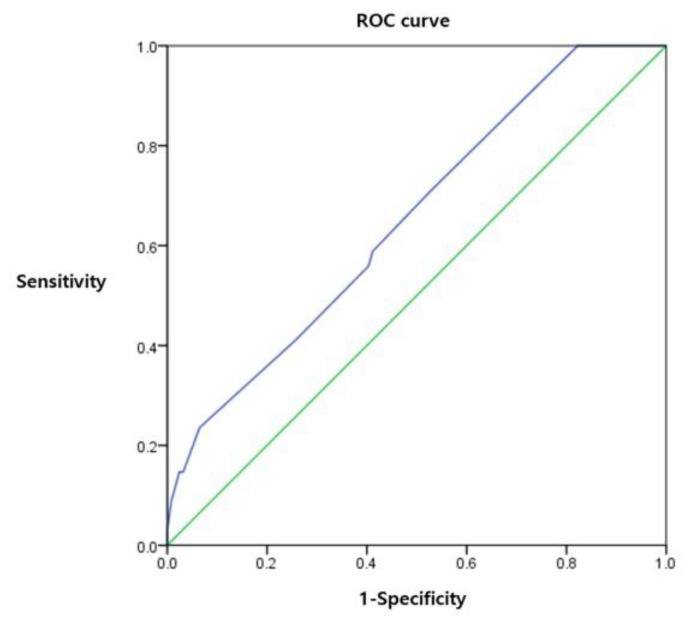
The receiver operating characteristic curve (ROC) was drawn to assess the ability of the COLDS score to predict perioperative respiratory adverse events (PRAEs).

**Table 1 medicina-58-01340-t001:** Logistic regression results for the occurrence of perioperative respiratory adverse events (PRAEs) regarding age, COLDS score, and ongoing URTI symptoms.

Independent Variables	OR	95% CI of OR	*p*-Value
(Constant)	0.040		0.002
Age (years)	0.855 *	0.772–0.947	0.003
COLDS score	1.223 *	1.022–1.465	0.028
Ongoing URTI symptoms	2.999 *	1.157–7.775	0.024

Adjusted R^2^ = 0.222, Hosmer and Lemeshow test *p* = 0.160, classification accuracy 81.0%. OR: odds ratio; CI: confidence interval. * *p* < 0.05.

**Table 2 medicina-58-01340-t002:** Demographic information of low COLDS group (Group L) and high COLDS group (Group H) after propensity matching.

Group	Group L (N = 69)	Group H (N = 69)	*p* Value
Sex (M/F)	43/26	43/26	1.000
Age (yr)	7.05 ± 5.03	7.96 ± 5.24	0.303
Height (cm)	113.75 ± 29.61	123.32 ± 32.91	0.080
Weight (kg)	27.10 ± 24.37	30.07 ± 18.88	0.425
Cough (Y/N)	35/26	47/20	0.133
Sputum (Y/N)	10/50	24/45	0.020
Rhinorrhea (Y/N)	36/26	27/40	0.044
Fever (Y/N)	16/44	18/49	0.980
Preoperative abnormal X-ray finding (Y/N)	0/69	8/61	0.006
Duration of URTI symptoms (day)	10.78 ± 10.49	9.06 ± 13.50	0.450
Active URTI (Y/N)	14/55	36/33	<0.001
URTI onset			<0.001
>1 month before surgery	6	0	
1 month~1 week before surgery	39	15	
1 day~1 week before surgery	10	18	
Within 1 day before surgery	14	36	
Type of surgery (major/minor)	18/51	21/48	0.571
Department of surgery (CS/GS/OS/PS/OL/OP/Others)	1/6/19/6/19/3/15	4/11/15/4/14/4/17	0.547
Anesthetic time (min)	126.22 ± 115.25	126.38 ± 93.24	0.993
Operation time (min)	77.41 ± 86.47	85.17 ± 83.13	0.591
Time in PACU (min)	59.70 ± 11.76	62.58 ± 10.12	0.134
C	1.22 ± 0.42	1.68 ± 0.96	<0.001
O	1.91 ± 0.78	4.43 ± 1.25	<0.001
L	1.01 ± 0.12	1.68 ± 1.47	<0.001
D	4.58 ± 1.18	5 ± 0	0.004
S	1.09 ± 0.28	1.45 ± 1.14	0.013
Total COLDS score	9.80 ± 1.38	14.25 ± 1.55	<0.001
Propensity score	0.44 ± 0.10	0.46 ± 0.11	0.257

URTI: upper respiratory tract infection; CS: cardiac surgery; GS: general surgery; OS: orthopedic surgery; PS: plastic surgery; OL: otorhinolaryngology; OP: ophthalmology; PACU: post anesthesia care unit; C: current symptoms; O: onset of URTI symptoms; L: lung disease; D: airway device; S: surgery type.

## Data Availability

The data in the study are accessible through correspondence.

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
