# Peer review of "Risk Assessment of Perioperative Respiratory Adverse Events and Validation of the COLDS Score in Children with Upper Respiratory Tract Infection"

_medicina, 2022, doi:10.3390/medicina58101340_

Round 1
Reviewer 1 Report
Congratulations to the authors for performing a complex chart review on a large group of patients. please see my specific comments below.
1. PRAE definitions vary widely depending what study we're looking. Clearly define what were the parameters considered as PRAE in this particular study and how each one was reviewed in the methods section. ex: one of the cited study mentions persistent coughing as PRAE, other study has spo2 less than 95% as PRAE, so clear definition is important. Also mention whether intraoperative bronchospasm, laryngospasm were included, and if so how they were tracked.
2. Page 1, line 40-41 and line 43-44 conflicts with each other. Airway sensitivity persists for 6 weeks, but recommendation is to postpone the case for 2 weeks. Please include further clarification with regards to why not wait 6 weeks with proper citation.
3. Line 43-44 needs to be rephrased. it is incomplete.
4. if the COLDS scoring was obtained by retrospective chart review of patient's history, how did the authors confirm that they were able to extract all the necessary data points to fulfill the COLDS score? please clarify in the methods section. how many missing data points on the COLDS scoring and how did the authors account for missing data. (example- did you have data on passive smoking, h/o RSV infection, intermittent or persistent asthma on every patient?)
4. Any information on preventive measures (prophylactic albuterol treatment), IV vs mask induction, use of rescue inhalers, succinylcholine, epinephrine to treat laryngospasm/bronchospasm episodes
5. line 132-133: please clarify all 34 cases of PRAEs happened in patients with active or recent URI? Does that mean none of the 6100 additional patients who underwent surgeries during the five year period had PRAEs? 34 cases of PRAEs over 5 yr periods time seems pretty low. Again, its important to clarify how PRAEs were defined in this study.
6. redundancy between table 2 and 3. Both convey the same information. I would do away with one of the tables.
Author Response
1. PRAE definitions vary widely depending what study we're looking. Clearly define what were the parameters considered as PRAE in this particular study and how each one was reviewed in the methods section. ex: one of the cited study mentions persistent coughing as PRAE, other study has spo2 less than 95% as PRAE, so clear definition is important. Also mention whether intraoperative bronchospasm, laryngospasm were included, and if so how they were tracked.
A: I fully agree with your opinion. We added the description for PRAE definition in the method section.
P3 L109-118: Since 2013, the in-department standards for URTI-related medical records have been strengthened, and postoperative respiratory events have been described in detail. The determination of PRAE was achieved by the presence of persistent cough, breath holding, hypoxemia (oxygen saturation <95% lasts more than 30 seconds.), laryngospasm or bronchospasm. Perioperative follow-up data were evaluated to confirm the occurrence of PRAEs using intraoperative anesthetic record and postoperative medical record in post-operative anesthetic unit (PACU) and ward. As mentioned above, the patients with current or recent URTI were documented in detail about their respiratory symptom and progression in intraoperative and postoperative period.
2. Page 1, line 40-41 and line 43-44 conflicts with each other. Airway sensitivity persists for 6 weeks, but recommendation is to postpone the case for 2 weeks. Please include further clarification with regards to why not wait 6 weeks with proper citation.
A: Current studies lack the evidence that 6 weeks of postponing reduces the risk of PRAE. Also the clinicions should also consider the risks of delaying surgery.
P2 L46-52: Therefore, when the children have severe URTI, delaying the surgery by at least 2 weeks after the active URTI symptoms is desirable [9,10]. In the cases of 2-6 weeks, the risk of airway hypersensitivity may remain. However, elective surgery may proceed considering the benefit of surgery. Postponing surgery can be stressful not only to the child and the parents but also to the surgeon and the hospital. Also, in emergency surgery, proceeding the surgery can outweigh the risk of PRAEs.
3. Line 43-44 needs to be rephrased. it is incomplete.
A: As your opinion, the manuscript was revised.
P1L43-P2L46: The incidence of PRAEs in children who have current or recent URTI was 25-29%, where-as that in children without URTI was 12% [1,8]. Zhang et al. [9] also reported that children who had URTI within 2 weeks showed significantly higher incidence of PRAEs.
4. if the COLDS scoring was obtained by retrospective chart review of patient's history, how did the authors confirm that they were able to extract all the necessary data points to fulfill the COLDS score? please clarify in the methods section. how many missing data points on the COLDS scoring and how did the authors account for missing data. (example- did you have data on passive smoking, h/o RSV infection, intermittent or persistent asthma on every patient?)
A: In 2013, we initially planned a prospective study, but the expected incidence of PRAE was low, and there were ethical and cost issues. Instead, the medical record was recorded in as much detail as possible to prepare for a retrospective study. Actually, the COLDS scoring was not difficult in the patients with preoperative URTI. In addition to the answer to Question 1, I revised manuscript. (In case of RSV infection, active or recent infection at preoperative consultation was recorded)
P3 L109-118: Since 2013, the in-department standards for URTI-related medical records have been strengthened, and postoperative respiratory events have been described in detail. The determination of PRAE was achieved by the presence of persistent cough, breath holding, hypoxemia (oxygen saturation <95% lasts more than 30 seconds.), laryngospasm or bronchospasm. Perioperative follow-up data were evaluated to confirm the occurrence of PRAEs using intraoperative anesthetic record and postoperative medical record in post-operative anesthetic unit (PACU) and ward. As mentioned above, the patients with current or recent URTI were documented in detail about their respiratory symptom and progression in intraoperative and postoperative period.
P3L123-124: Pre-operative summary also includes underlying diseases of the patient, asthma or allergy and passive smoking history.
4. Any information on preventive measures (prophylactic albuterol treatment), IV vs mask induction, use of rescue inhalers, succinylcholine, epinephrine to treat laryngospasm/bronchospasm episodes
A: Routine prophylactic albuterol treatment was not performed. All cases used IV induction (thiopental or propofol). There was no case for succinylcholine use. 7 used inhaler in PACU. We used steroid and epinephrine to treat laryngospasm cases. We added the specific description in the result section.
P3 L145-148: 14 patients showed persistent cough, 10 showed breath holding, 27 showed hypoxemia, and 2 showed laryngospasm. Seven patients used steroid or inhaler in PACU. One patient showed newly occurred pneumonic infiltration in the postoperative chest x-ray.
5. line 132-133: please clarify all 34 cases of PRAEs happened in patients with active or recent URI? Does that mean none of the 6100 additional patients who underwent surgeries during the five year period had PRAEs? 34 cases of PRAEs over 5 yr periods time seems pretty low. Again, its important to clarify how PRAEs were defined in this study.
A: Sorry for making you confused. 34 of PRAEs were reported in the patients with active or recent URTI. The specific records of 158 people of preoperative URTI can be believed to be accurate. For the remaining 6,100 patients, PRAE and COLDS scoring were somewhat inaccurate, so their result was not reported. I revised the manuscript.
P3L144-148: A total of 158 patients had current or recent URTI, and 34 cases (21.5%) of PRAEs were reported among the pediatric patients with current or recent URTI.
6. redundancy between table 2 and 3. Both convey the same information. I would do away with one of the tables.
A: Thank you for your advice. Table 2 was now moved to supplementary table 3.
Thank you again for your considerate review. Your opinion was very useful in promoting and revising the paper.

Reviewer 2 Report
The Authors presented the results of a single-center retrospective study on patients aged below 18 years, undergoing elective surgery across a time frame of 5 years. The study investigated multiple risk factors associated with perioperative respiratory adverse events, includig those evaluated by the COLDS scoring system to validate it.
The research design is appropriate, results are adequately discussed, and limitations are thoroughly adressed.
However, I strongly recommend to provide more recent and updated references in the introduction and discussion sections.
Also, the caption for Figure 1 (b) is not clear, a more detailed description would be beneficial for a clearer understanding of the data presented.
Author Response
Answer: Thank you for your kind review. As your opinion, references were added in the introduction and discussion sections.
- Berlie C.W., Yaregal M.D. Incidence and associated factors of laryngospasm among pediatric patients who underwent surgery under general anesthesia, in university of Gondar compressive specialized hospital, Northwest Ethiopia, 2019: A Cross-sectional study. Anesthesiol Res Pract 2020, eCollection 2020, 3706106, http://dx.doi.org/10.1155/2020/3706106
- Regli, A.; Becke, K.; von Ungern-Sternberg, B.S. An update on the perioperative management of children with upper respiratory tract infections. Curr. Opin. Anaesthesiol. 2017, 30, 362-367, https://dx.doi.org/10.1097/aco.0000000000000460.
- De Carvalho A.L.R., Vital R.B., de Lira C.C.S., Margo I.B., Sato P.T.S., Lima L.H.N., Braz L.G., Modolo N.S.P. Laryngeal mask airway versus other airway devices for anesthesia in children with an upper respiratory tract infection: A systematic review and meta-analysis of respiratory complications. Anesth Analg 2018, 127, 941-950, http://dx.doi.org/10.1213/ANE.0000000000003674.
- Michel F, Vacher T, Julien-Marsollier F, Dadure C, Aubineau J.V., Sabourdin N, Woodey E, Orliaguet G, Brasher C, Dahmani S. Peri-operative respiratory adverse evenets in children with upper respiratory tract infections allowed to proceed with ananesthsia: A Frech national cohort study. Eur J Anaethesiol 2018, 35, 919-928, http://dx.doi.org/10.1097/EJA.000000000000875.
Also, the captions for figure 1(b) were revised for your advice.
Figure 1. The graph depicting occurrence of perioperative respiratory adverse events (PRAEs) according to ‘COLDS’ score. (a) Number of patients who suffered PRAEs according to ‘COLDS’ score. (b) Description of each COLDS category in the patients with PRAEs. The five components of the COLDS score did not contribute equally to the total score. Total COLDS scores were contributed in the order of D, O, and the others. *p<0.05 compared to “C”, “L” or “S”, †p<0.05 com-pared to “O”.
Thank you again for your valuable advice. It was helpful to revise and promote the paper.

Reviewer 3 Report
I read with interest this manuscript submitted for publication in Medicina.
Here are my comments and suggestions.
1. The article at first reading seems difficult to assimilate. I strongly suggest you try to make it more reader-friendly.
2.In general, references should be expanded.
3. In addition, moderate English changes are required.
4. An abbreviation sections is necessary.
Author Response
Answer: I am very sorry to make you confused in manuscript review. I revised to clarify the flow of the study conduction and to rearrange the sentences.
Changes are marked in red. A certificate for English correction has also been added.
Four references were added for your advice.
- Berlie C.W., Yaregal M.D. Incidence and associated factors of laryngospasm among pediatric patients who underwent surgery under general anesthesia, in university of Gondar compressive specialized hospital, Northwest Ethiopia, 2019: A Cross-sectional study. Anesthesiol Res Pract 2020, eCollection 2020, 3706106, http://dx.doi.org/10.1155/2020/3706106
- Regli, A.; Becke, K.; von Ungern-Sternberg, B.S. An update on the perioperative management of children with upper respiratory tract infections. Curr. Opin. Anaesthesiol. 2017, 30, 362-367, https://dx.doi.org/10.1097/aco.0000000000000460.
- De Carvalho A.L.R., Vital R.B., de Lira C.C.S., Margo I.B., Sato P.T.S., Lima L.H.N., Braz L.G., Modolo N.S.P. Laryngeal mask airway versus other airway devices for anesthesia in children with an upper respiratory tract infection: A systematic review and meta-analysis of respiratory complications. Anesth Analg 2018, 127, 941-950, http://dx.doi.org/10.1213/ANE.0000000000003674.
- Michel F, Vacher T, Julien-Marsollier F, Dadure C, Aubineau J.V., Sabourdin N, Woodey E, Orliaguet G, Brasher C, Dahmani S. Peri-operative respiratory adverse evenets in children with upper respiratory tract infections allowed to proceed with ananesthsia: A Frech national cohort study. Eur J Anaethesiol 2018, 35, 919-928, http://dx.doi.org/10.1097/EJA.000000000000875.
Abbreviation section was added for your advice.
Abbreviation: CS: cardiac surgery; GS; general surgery; OP; ophthalmology; OS: or-thopedic surgery; OL: otorhinolaryngology; PRAEs: perioperative respiratory adverse events, PS: plastic surgery, PACU: post anesthesia unit, URTI: upper respiratory tract infection.
Thank you again for your review. Your opinions were useful in promoting and revision the paper.

Round 2
Reviewer 1 Report
Thank you for making the suggested changes. The article reads well and adds value to the existing knowledge of pediatric URIs and anesthesia.
Reviewer 3 Report
The authors answered to all my comments . The manuscript can be published in the actual form.